# BFV-Based Homomorphic Encryption for Privacy-Preserving CNN Models

**Febrianti Wibawa [1], Ferhat Ozgur Catak [1,*], Salih Sarp [2], Murat Kuzlu [3]**

[1] Department of Electrical Engineering and Computer Science, University of Stavanger, 4021 Rogaland, Norway; f.febrianti@gmail.com

[2] Department of Electrical and Computer Engineering, Virginia Commonwealth University, Richmond, VA 23284, USA; sarps@vcu.edu

[3] Batten College of Engineering and Technology, Old Dominion University, Norfolk, VA 23529, USA; mkuzlu@odu.edu

[*] Correspondence: f.ozgur.catak@uis.no

**Abstract:** Medical data is frequently quite sensitive in terms of data privacy and security. Federated learning has been used to increase the privacy and security of medical data, which is a sort of machine learning technique. The training data is disseminated across numerous machines in federated learning, and the learning process is collaborative. There are numerous privacy attacks on deep learning (DL) models that attackers can use to obtain sensitive information. As a result, the DL model should be safeguarded from adversarial attacks, particularly in medical data applications. Homomorphic encryption-based model security from the adversarial collaborator is one of the answers to this challenge. Using homomorphic encryption, this research presents a privacy-preserving federated learning system for medical data. The proposed technique employs a secure multi-party computation protocol to safeguard the deep learning model from adversaries. The proposed approach is tested in terms of model performance using a real-world medical dataset in this paper.

**Keywords:** medical data; homomorphic encryption; deep learning





## 1. Introduction

Machine learning (ML) is a widely used technique in almost all fields, where a computer system can learn from data to improve its performance. This technique is widely used in many application areas such as image recognition, natural language processing, and machine translation. Federated learning is a machine learning technique where the training data is distributed across multiple machines, and the learning process is performed in a collaborative manner [1]. This technique can be used to improve the privacy and security of medical data [2].

Medical data is highly sensitive and is often subject to data privacy and security concerns [3]. For example, a person's health information is often confidential and can be used to identify the person. Thus, it is essential to protect the privacy and security of medical data. The Health Insurance Portability and Accountability Act (HIPAA) (US Department of Health and Human Services, 2014) and General Data Protection Regulation (GDPR) (The European Union, 2018) strictly mandate personal health information privacy. There are various methods to safeguard the private information. Federated learning is one of the techniques that can be utilized for the protection of sensitive data during multi-party computation tasks. This technique can be used to improve the privacy and security of medical data by preventing the data from being centralized and vulnerable.

Keeping the data local is not sufficient for the security of the data and the ML model. However, there are several privacy attacks on deep learning models to get the private data [4,5]. For example, the attackers can use the gradient information of the deep learning

model to get the sensitive information. Thus, the deep learning model itself should be protected from the adversaries as well. One of the solutions for this problem is homomorphic encryption-based model protection from the adversary collaborator. Homomorphic encryption is a technique where the data can be encrypted, and the operations can be performed on the encrypted data [6]. This technique can be used to protect the deep learning model from the adversaries.

This paper proposes a privacy-preserving federated learning algorithm based on a convolutional neural network (CNN) for medical data using homomorphic encryption. The proposed algorithm uses a secure multi-party computation protocol to protect the deep learning model from the adversaries. We evaluate the proposed algorithm using a real-world medical dataset and show that the proposed algorithm can protect the deep learning model from the adversaries. We limited our work to binary classification. In our subsequent work, we plan to use multi-class approaches in the literature [7,8].

The main contributions of this paper are as follows:

- A recent European Data Protection Board (EDPB) Public Consultation stated the use of Secure Multi-Party Computation as an additional measure to the General Data Protection Regulation's (GDPR) Article 46 transfer tools. Here, we provide a method to implement practically secure multi-party computation in federated learning to improve the privacy and security of medical data (https://edpb.europa.eu/sites/default/files/webform/public_consultation_reply/inpher-_edpb_supplementary_measures_comment.pdf (27 June 2022)).
- A homomorphic encryption-based federated learning algorithm is proposed to protect the confidentiality of the sensitive medical data.
- A secure multi-party computation protocol is proposed to protect the deep learning models from the adversaries.
- A real-world medical dataset is used to evaluate the proposed algorithm. The experimental results show that the proposed algorithm can protect the deep learning model from the adversaries.

The rest of the paper is organized as follows: Section 2 describes related work, while Section 3 describes the preliminaries, including homomorphic encryption and federated learning. Section 4 provides detailed information of the proposed system model. Sections 5 and 6 discuss the results, while Section 7 concludes the paper.

## 2. Related Work

Data-driven ML models provide unprecedented opportunities for healthcare with the use of sensitive health data. These models are trained locally to protect the sensitive health data. However, it is difficult to build robust models without diverse and large datasets utilizing the full spectrum of health concerns. Prior proposed works to overcome this problem include federated learning techniques. For instance, the studies [9–11] reviewed the current applications and technical considerations of the federated learning technique to preserve the sensitive biomedical data. The impact of federated learning is examined through the stakeholders, such as patients, clinicians, healthcare facilities, and manufacturers. In another study, the authors in [12] utilized federated learning systems for brain tumor segmentation on the BraTS dataset, which consists of magnetic resonance imaging brain scans. The results show that performance is decreased by privacy protection costs. The same BraTS dataset is used in [13] to compare three collaborative training techniques, i.e., federated learning, institutional incremental learning (IIL), and cyclic institutional learning (CIIL). In IIL and CIIL, institutions train a shared model successively, where CIIL adds a cycling loop through organizations. The results indicate that federated learning achieves similar Dice scores to that of models trained by sharing data. It outperforms the IIL and CIIL methods since these methods suffer from catastrophic forgetting and complexity.

Medical data is also safeguarded by encryption techniques such as homomorphic encryption. In [14], authors propose an online secure multi-party computation sharing patient information to hospitals using homomorphic encryption. Bocu et al. [15] proposed a homomorphic encryption model that is integrated with a personal health information sys-

tem utilizing heart rate data. The results indicate that the described technique successfully addressed the requirements for secure data processing for the 500 patients with expected storage and network challenges. In another study by Wang et al. [16], a data division scheme based on homomorphic encryption for wireless sensor networks was proposed. The results show that there is a trade-off between resources and data security. In [17], the applicability of homomorphic encryption is shown by measuring the vitals of the patients with a lightweight encryption scheme. Sensor data such as respiration and heart rate are encrypted using homomorphic encryption before transmitting to the non-trusting third party, while encryption takes place only in a medical facility. The study in [18] developed an IoT-based architecture with homomorphic encryption to combat data loss and spoofing attacks for chronic disease monitoring. The results suggest that homomorphic encryption provides cost-effective and straightforward protection of the sensitive health information. Blockchain technologies are also utilized in cooperation with homomorphic encryption for the security of medical data. Authors in [19] proposed a practical pandemic infection tracking tool using homomorphic encryption and blockchain technologies in intelligent transportation systems using automatic healthcare monitoring. In another study, Ali et al. [20] developed a search-able distributed medical database on a blockchain using homomorphic encryption. The increase need to secure sensitive information leads to the use of various techniques together. In the scope of this study, a multi-party computation tool using federated learning with homomorphic encryption is developed and analyzed.

### 3. Preliminaries

*3.1. Homomorphic Encryption*

The definition of the homomorphic encryption (HE) scheme is given in [21] as follows:

**Definition 1** (**Homomorphic Encryption**). *A family of schemes $\{\mathcal{E}_k\}_{k \in \mathbb{Z}_+}$ is said to be homomorphic with respect to an operator $\circ$ if there exist decryption algorithms $\{\mathcal{D}_k\}_{k \in \mathbb{Z}_+}$ such that for any two ciphertexts $c_1, c_2 \in \mathcal{C}$, the following equality is satisfied:*

$$\mathcal{D}_k(\mathcal{E}_k(m_1, r_1) \circ \mathcal{E}_k(m_2, r_2)) = m_1 \circ m_2, \ \forall m_1, m_2 \in \mathcal{M}, \tag{1}$$

*where $r_1, r_2 \in \mathcal{R}$ are the corresponding randomness.*

A homomorphic encryption scheme is a pair of algorithms, Enc and Dec, with the following properties:

- Enc takes as input a plaintext $m \in \mathbb{Z}_N$, and outputs a ciphertext $c$ such that $c$ is a homomorphic image of $m$, i.e., Dec($c$) = $m$;
- Dec takes as input a ciphertext $c$, and outputs a plaintext $m$ such that $m$ is a homomorphic image of $c$;
- Enc and Dec are computationally efficient.

There are two types of homomorphic encryption: additively homomorphic and multiplicatively homomorphic.

Additively homomorphic encryptionconsists of a pair of algorithms Enc and Dec such that, for all $m_1, m_2 \in \mathbb{Z}_N$, $c_1 = \text{Enc}(m_1)$, $c_2 = \text{Enc}(m_2)$, and $c_3 = c_1 + c_2$, we have Dec($c_3$) = $m_1 + m_2$.

Multiplicatively homomorphic encryption consists of a pair of algorithms Enc and Dec such that, for all $m_1, m_2 \in \mathbb{Z}_N$, $c_1 = \text{Enc}(m_1)$, $c_2 = \text{Enc}(m_2)$, and $c_3 = c_1 c_2$, we have Dec($c_3$) = $m_1 m_2$.

Partially homomorphic encryption is a variant of homomorphic encryption where homomorphism is only partially supported, i.e., the encryption scheme is homomorphic for some operations while not homomorphic for others.

Somewhat homomorphic encryption is a variant of fully homomorphic encryption where homomorphism is only limited supported, i.e., the encryption scheme is homomorphic for all operations for a limited number of operations.

Fully homomorphic encryption (FHE) is a variant of homomorphic encryption which allows for homomorphism over all functions, i.e., the encryption scheme is homomorphic for all operations. In other words, an FHE scheme consists of a pair of algorithms Enc and Dec such that, for all $m_1, m_2 \in \mathbb{Z}_N$, $c_1 = \mathsf{Enc}(m_1)$, $c_2 = \mathsf{Enc}(m_2)$, and $c_3 = c_1 c_2$, we have $\mathsf{Dec}(c_3) = m_1 m_2$.

Table 1 shows a summary of the major homomorphic encryption schemes.

**Table 1.** Major homomorphic encryption schemes.

| Scheme | Key-Size | Additive/Multiplicative | Partially/Somewhat/Fully |
|---|---|---|---|
| Paillier | 2048 bits | Additive | Partially |
| ElGamal | 1024 bits | Additive | Partially |
| BFV | 2048 bits | Additive, Multiplicative | Somewhat |
| CKKS | 2048 bits | Multiplicative | Somewhat |
| FV | 2048 bits | Multiplicative | Somewhat |

*3.2. Brakerski–Fan–Vercauteren (BFV) Scheme*

Since the work of Brakerski, Fan, and Vercauteren (BFV), the somewhat homomorphic encryption (SHE) scheme has become one of the most important research topics in cryptography. In this section, we give the definition of this scheme.

**Definition 2 (BFV scheme).** *An SHE scheme $\mathcal{E}$ is said to be in the BFV family of schemes if it consists of the following three algorithms:*

- *Key generation algorithm: It takes the security parameter k as input, and outputs a public key pk and a secret key sk.*
- *Encryption algorithm: It takes the message $m \in \mathcal{M}$, a public key pk, and a randomness $r \in \mathcal{R}$ as inputs, and outputs a ciphertext $c \in \mathcal{C}$.*
- *Decryption algorithm: It takes a ciphertext $c \in \mathcal{C}$, a secret key sk, and an integer $i \in \mathbb{Z}_+$ as inputs, and outputs a message $m \in \mathcal{M}$.*

**Remark 1.** *In the above definition, the integer i is called the* decryption index. *It is introduced to allow for efficient decryption of ciphertexts that are the result of homomorphic operations. For example, when the ciphertext $c_1$ is the result of homomorphic operations on ciphertexts $c_2$ and $c_3$, that is, $c_1 = c_2 \circ c_3$, then $c_1$ can be decrypted by taking the decryption index $i = 2$.*

In the following, we give a brief description of the BFV scheme.

The key generation algorithm of the BFV scheme consists of the following two steps.

1. Let $t$ be the security parameter. For a positive integer $t$, define a number $n = \lfloor b(t) \rfloor$ and a positive integer $p$ where $b : \mathbb{Z}_+ \to \mathbb{Z}_+$ is a polynomial, and $p$ is a prime number satisfying $p > 2^n$.
2. Let $d$ be a positive integer such that $d < p$. Choose a monic polynomial $f(x)$ of degree $d$ with $f(x) \equiv x - \tilde{a} \pmod{p}$ for some $\tilde{a} \in \mathbb{Z}_p$. Let $T(x) = x^n f(x) \pmod{p}$. Choose a quadratic nonresidue $b$ of $\mathbb{Z}_p$, and let $L(x) = T(x) b x^{\frac{n}{2}} \pmod{p}$.

Let $q = 2nL(0)$. The secret key $sk$ is chosen to be a nonnegative integer $s$ less than $q$. The public key $pk$ is chosen to be the sequence $(p, T(x), L(x), n, q)$.

The encryption algorithm of the BFV scheme consists of the following three steps.

1. Let $pk = (p, T(x), L(x), n, q)$ be the public key. Choose a random polynomial $R(x) \in \mathbb{Z}_p[x]$ of degree less than $d$.
2. Given a message $m \in \mathbb{Z}_p$, compute $u(x) = m + \frac{1}{2} R(x)^2 L(x)^{-1} \pmod{p}$.
3. Choose a random integer $\tilde{t} \in \mathbb{Z}_p$, and output the ciphertext $c = (\tilde{t}, u(x))$.

The decryption algorithm of the BFV scheme consists of the following two steps.

1. Let $sk = s$ be the secret key. Compute $v(x) = L(x)^{-1}(s + \frac{1}{2} T(x)^2 b^{-1} x^n) \pmod{p}$.

2.  Given a ciphertext $c = (\tilde{t}, u(x))$, compute $m = u(x) - \frac{1}{2}v(x)^2 \pmod{p}$.

**Remark 2.** *In the BFV scheme, the message space is $\mathcal{M} = \mathbb{Z}_p$.*

### 3.2.1. Homomorphic Operations
### Additive Homomorphism

In the BFV scheme, the additive homomorphism is defined as follows:

**Definition 3 (Additive homomorphism).** *Let $c_1 = (\tilde{t}_1, u_1(x))$ and $c_2 = (\tilde{t}_2, u_2(x))$ be two ciphertexts. The additive homomorphism is defined to be the ciphertext $c_1 + c_2 = (\tilde{t}_1 + \tilde{t}_2, u_1(x) + u_2(x))$.*

**Remark 3.** *In the BFV scheme, the standard polynomial addition algorithm implements the additive homomorphism.*

### Multiplicative Homomorphism

In the BFV scheme, the multiplicative homomorphism is defined as follows:

**Definition 4 (Multiplicative homomorphism).** *Let $c_1 = (\tilde{t}_1, u_1(x))$ be a ciphertext and $m \in \mathbb{Z}_p$ be a message. The multiplicative homomorphism is defined to be the ciphertext $c_1 \cdot m = (\tilde{t}_1 \cdot m, u_1(x) \cdot m)$.*

**Remark 4.** *In the BFV scheme, the standard polynomial multiplication algorithm implements the multiplicative homomorphism.*

**Remark 5.** *The multiplicative homomorphism is sometimes called the "plaintext multiplication" or the "scalar multiplication".*

### 3.2.2. Relinearization

Relinearization is a homomorphic operation used in the BFV scheme to reduce the number of ciphertexts generated by homomorphic operations. In the following, we give the definition of this operation.

**Definition 5 (Relinearization).** *Let $c_1 = (\tilde{t}_1, u_1(x))$ and $c_2 = (\tilde{t}_2, u_2(x))$ be two ciphertexts. The relinearization homomorphism is defined to be the ciphertext $c_1 + c_2 T(x) = (\tilde{t}_1 + \tilde{t}_2 T(x), u_1(x) + u_2(x)T(x))$.*

**Remark 6.** *In the BFV scheme, the relinearization homomorphism is implemented by the standard polynomial addition and multiplication algorithms.*

### 3.2.3. Rotation

Rotation is a homomorphic operation used in the BFV scheme to implement the power operation efficiently. It can be used to implement a large class of homomorphic operations on encrypted data. In the following, we give the definition of this operation.

**Definition 6 (Rotation).** *Let $c = (\tilde{t}, u(x))$ be a ciphertext. The rotation homomorphism is defined to be the ciphertext $c^r = (\tilde{t}, u(x)^r)$, where $r$ is an integer.*

**Remark 7.** *In the BFV scheme, the rotation homomorphism is implemented by the standard polynomial multiplication algorithm.*

**Remark 8.** *The rotation is sometimes called the "power operation".*

### 3.3. Federated Learning

In this section, we briefly describe the federated learning (FL) framework. We refer to [22,23] for more details.

**Definition 7 (FL model).** *Let N be a positive integer, and $\mathcal{X}$ be a probability space. Let m be a positive integer such that $m < N$, and $\mathcal{P} = \{p_1, p_2, \ldots, p_m\}$ be a collection of random variables on $\mathcal{X}$ with $p_i \in \mathcal{L}_1(\mathcal{X})$ for $i = 1, 2, \ldots, m$. The FL model consists of the following four algorithms:*

- *Initialization algorithm: It takes the security parameter k as input, and outputs the global model $w_0 \in \mathbb{R}^n$, where n is the number of free parameters in $w_0$.*
- *Local training algorithm: It takes the global model $w_t \in \mathbb{R}^n$, a local dataset $D_i \in \mathcal{D}$, and a positive integer t as inputs, and outputs a local model $w_{t+1}^i \in \mathbb{R}^n$.*
- *Upload algorithm: It takes the local model $w_t^i \in \mathbb{R}^n$, and a positive integer t as inputs, and outputs a vector $v_t^i \in \mathbb{R}^n$.*
- *Aggregation algorithm: It takes a set of vectors $v_t^i \in \mathbb{R}^n$, and a positive integer t as inputs, and outputs the global model $w_{t+1} \in \mathbb{R}^n$.*

In the above definition, the integer $t$ is called the *training round*. The global model $w_t$ is a function of the training round $t$. The global model $w_t$ is trained by the local models $w_t^i$, which are trained on the local datasets $D_i$. The global model $w_t$ is trained on the aggregated dataset $\cup_{i=1}^m D_i$. The global model $w_t$ is initialized to be the global model $w_0$.

**Remark 9.** *In the FL model, the local training algorithm, upload algorithm, and aggregation algorithm can be implemented by any machine learning algorithm.*

**Remark 10.** *The global model $w_t$ can be trained on the aggregated dataset $\cup_{i=1}^m D_i$ using any machine learning algorithm.*

**Remark 11.** *In the FL model, the global model $w_t$ is shared among all the participating clients, and the local models $w_t^i$ are not shared among the clients.*

### 4. System Model

This section gives a high-level system overview of the proposed BFV crypto-scheme-based privacy-preserving federated learning COVID-19 detection training method. The proposed privacy-preserving scheme is a two-phase approach: (1) local model training at each client and (2) encrypted model weight aggregation at the server. In the local model training phase, each client builds their local CNN-based DL model using their local electronic health record dataset. The clients encrypt the model weights matrix using the public key. In the second step, the server aggregates all clients' encrypted weight matrices and sends the final matrix to the clients. Each client decrypts the aggregated encrypted weight matrix to update the model weights of their DL model. Figure 1 shows the system overview.

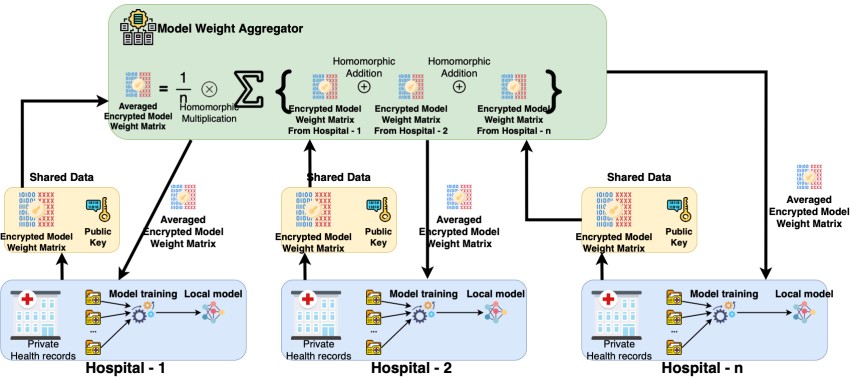

**Figure 1.** Overall system overview of the proposed method.

Figure 2 shows the CNN-based COVID-19 detection model used in the experiments.

**Figure 2.** CNN-based COVID-19 detection model.

### 4.1. Notations

- Boldface lowercase letters show the vectors (e.g., $\mathbf{x}$);
- $[\![W]\!]$ shows the ciphertext of a matrix $W$;
- $\oplus$ shows the homomorphic encryption-based addition, $\otimes$ homomorphic encryption-based multiplication;
- $(key_{pub}, key_{priv})$ shows public/private key pairs.

### 4.2. Client Initialization

Algorithm 1 shows the overall process in the initialization phase. Each client trains the local classifier, $h_i$, with their private dataset $\mathcal{D}_i$. The trained model's weight matrix, $W$, is encrypted, $[\![W]\!]$, and shared with the server

---

**Algorithm 1** Model training in each client

---

**Require:** The dataset at client $c$: $\mathcal{D}_c = \{(\mathbf{x}, y) | \mathbf{x} \in \mathbb{R}^m, y \in \mathbb{R}\}_{i=0}^m$, public key: $Key_{pub}$
1: $X_{train}, X_{test}, \mathbf{y}_{train}, \mathbf{y}_{test} \leftarrow train\_test\_split(\mathcal{D})$
2: $h \leftarrow global\_model$
3: $h.fit(X_{train}, \mathbf{y}_{train})$
4: $W \leftarrow \varnothing$ // Create an empty matrix for the encrypted layer weights
5: **for each** $layer \in h$ **do**
6:     $[\![W]\!] \leftarrow encrypt\_fractional(layer.weights, key_{pub})$ // Encrypt the layer weights (layer.weights $\in \mathbb{R}^m$) with public key.
7: **end for**
8: **Return** $[\![W]\!]$ // The encrypted weight matrix

---

### 4.3. Model Aggregation

The server collects all encrypted weight matrices, $\{[\![W]\!]_0, \cdots, [\![W]\!]_c\}$, from the clients. It calculates the average weight value of each neuron in the encrypted domain. Algorithm 2 shows the overall process in the aggregation phase.

---
**Algorithm 2** Model aggregation at the server

---
**Require:** public key: $Key_{pub}$, the number of clients: $c$, client model weights: $H = \{[\![W]\!]_i\}_{i=0}^c$
1: $[\![W]\!]_{aggr} \leftarrow \varnothing$
2: **for each** $h \in H$ **do**
3: 　　**for each** $[\![row]\!] \in h$ **do**
4: 　　　　$[\![W]\!]_{aggr} \leftarrow [\![W]\!]_{aggr} \oplus [\![row]\!]$ // *Homomorphic addition*
5: 　　**end for**
6: **end for**
7: **for each** $[\![row]\!] \in [\![W]\!]_{aggr}$ **do**
8: 　　$[\![row]\!] \leftarrow [\![row]\!] \otimes c^{-1}$ // *Homomorphic multiplication.*
9: **end for**
10: **Return** $[\![W]\!]_{aggr}$ // *Return the aggregated weight matrix in the encrypted domain*

---

### 4.4. Client Decryption

The last step is client decryption in which each client decrypts the aggregated and encrypted weight matrix, $[\![W]\!]_{aggr}$, and updates their local model, $h$. Algorithm 3 shows the overall process in the client decryption phase.

---
**Algorithm 3** Client decryption

---
**Require:** private key: $Key_{priv}$, encrypted aggregated weights: $[\![W]\!]_{aggr}$
1: $h \leftarrow global\_model$
2: **for each** $layer \in h$ **do**
3: 　　$[\![row]\!] \leftarrow [\![W]\!]_{aggr}(layer)$ // *Get the corresponding row for layer*
4: 　　$layer \leftarrow decrypt\_fractional([\![row]\!], key_{priv})$ // *Decrypt the row and update the layer weights*
5: **end for**
6: $h.save\_model(global\_model)$ // *Save the aggregated model as global_model at client.*

---

## 5. Results

### 5.1. Dataset

In this work, a COVID-19 radiography dataset collected in previous works related to a COVID-19 detection model [24,25] was used. The dataset contains X-ray lung images with four different classifications, which are COVID, Lung_OPACITY, Normal, and Viral Pneumonia. In this work, we utilized two classifications, which are COVID and Normal, focusing only on the COVID-19 detection machine learning process.

Samples of the dataset are depicted in Figures 3 and 4.

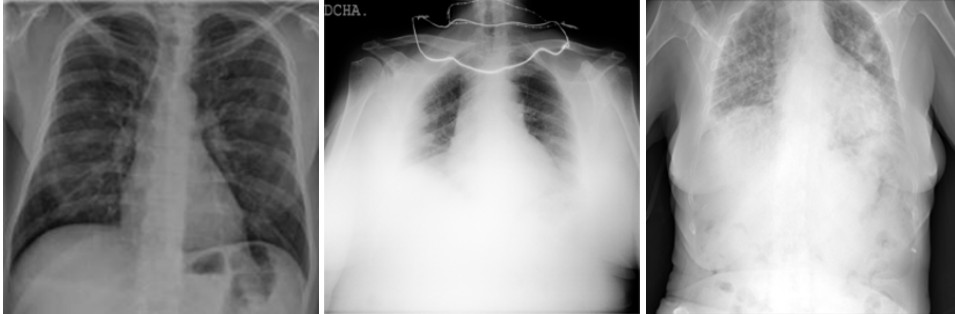

**Figure 3.** COVID-19 positive X-ray image dataset samples.

From the original dataset, we obtained the first 1000 records from each classification, with 80% of the sample used for the training set and the remaining 20% as the test set. The training dataset was further split with 10% of the dataset as the train-validation dataset.

We obtained 1000 records for each classification with 800 records used as the training dataset and 200 records used as the test dataset.

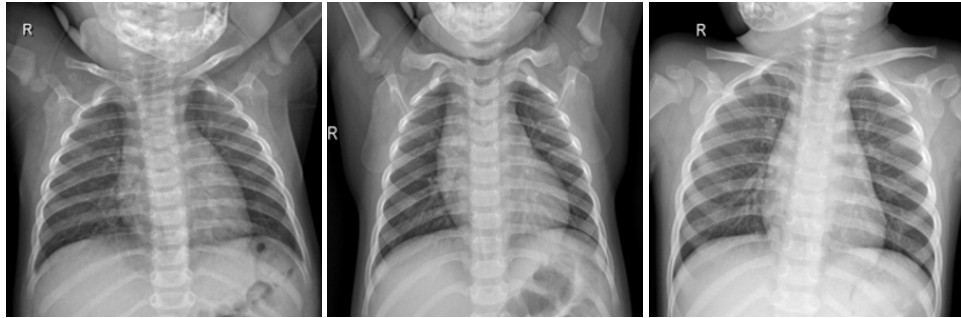

**Figure 4.** COVID-19 negative X-ray image dataset samples.

### 5.2. Preprocessing

Data preprocessing performed in the work consists of data augmentation and rescaling for the training dataset, while only data rescaling was applied for the test dataset. Data augmentation was necessary in order to provide data variety in the training dataset. The dataset was rescaled by multiplying the pixel value with 1/255. This was aimed to transform the pixel value range from [0, 255] to [0, 1] so that pixels were treated in the same way.

## 6. Implementation

### 6.1. Experimental Setup

Table 2 shows the CNN model used to predict COVID-19 detection.

**Table 2.** Model summary.

| Layer (Type) | Output | Shape | No. of Parameters |
| --- | --- | --- | --- |
| conv2d | (Conv2D) | (None, 254, 254, 32) | 896 |
| max_pooling2d | (MaxPooling2D) | (None, 127, 127, 32) | 0 |
| conv2d_1 | (Conv2D) | (None, 125, 125, 32) | 9248 |
| max_pooling2d_1 | (MaxPooling 2D) | (None, 62, 62, 32) | 0 |
| conv2d_2 | (Conv2D) | (None, 60, 60, 32) | 9248 |
| max_pooling2d_2 | (MaxPooling 2D) | (None, 30, 30, 32) | 0 |
| conv2d_3 | (Conv2D) | (None, 28, 28, 64) | 18,496 |
| max_pooling2d_3 | (MaxPooling 2D) | (None, 14, 14, 64) | 0 |
| conv2d_4 | (Conv2D) | (None, 12, 12, 64) | 36,928 |
| max_pooling2d_4 | (MaxPooling 2D) | (None, 6, 6, 64) | 0 |
| conv2d_5 | (Conv2D) | (None, 4, 4, 128) | 73,856 |
| max_pooling2d_5 | (MaxPooling 2D) | (None, 2, 2, 128) | 0 |
| flatten | (Flatten) | (None, 512) | 0 |
| dense | (Dense) | (None, 128) | 65,664 |
| dense_1 | (Dense) | (None, 64) | 8256 |
| dense_2 | (Dense) | (None, 2) | 130 |

Total params: 222,722. Trainable params: 222,722. Non-trainable params: 0.

The implementation was developed with Python 3.8.8, using existing libraries. There were standard libraries used, such as Keras and Tensorflow, which were used in the machine learning processes; Numpy, which was used to process weight arrays and data structures; pickle, to serialize exported weights; and most importantly, Pyfhel, which is used for homomorphic encryption. Pyfhel [26] is basically a Python wrapper for Microsoft SEAL, which provides the same functionalities as the Microsoft Simple Encrypted Arithmetic Library (SEAL).

Microsoft SEAL is a homomorphic encryption library developed by Microsoft, which was released in 2015. The SEAL library implements both Brakerski–Fan–Vercauteren (BFV) [27,28] and Cheon–Kim–Kim–Song (CKKS) [29] homomorphic encryption schemes and provides standard SHE functions starting from encoding, key generation, encryption, decryption, additive, multiplicative, and relinearization functions.

In the Pyfhel library implementation, we implemented the BFV scheme and used pre-defined default values in the HE context parameters, but with exception to parameter *sec*. Parameter *sec* is used to determine the bit-wise security level provided. At the time of writing, there are two possible values—128 and 192. We experimented with these values and observed the overall model performance in terms of time and accuracy. The parameter *sec* determines the length of coefficient modulus $(q)$ based on polynomial degree $(n)$ parameter, as described in the below Table 3 [30].

**Table 3.** Default pairs $(n, q)$ for 128-bit and 192-bit security levels.

| | Bit-Length of Default $q$. | |
|---|---|---|
| **n** | **128-bit Security** | **192-bit Security** |
| 1024 | 27 | 19 |
| 2048 | 54 | 37 |
| 4096 | 109 | 75 |
| 8192 | 218 | 152 |
| 16,384 | 438 | 300 |
| 32,768 | 881 | 600 |

We have implemented our proposed protocols and the classifier training phase in Python by using the Keras/Tensorflow libraries for the model building and the Microsoft SEAL library for the somewhat homomorphic encryption implementation. To show the training phase time performance of the proposed protocols, we tested the COVID-19 X-ray scans public dataset with different numbers of clients and the ciphertext modulus, $q = \{128, 192\}$, which determines how much noise can accumulate before decryption fails. Table 4 shows the dataset details.

**Table 4.** Dataset description.

| Dataset | Rows | Label |
|---|---|---|
| Training | 800 | Negative |
| | 800 | Positive |
| Test | 200 | Negative |
| | 200 | Positive |

The dataset is arbitrarily partitioned among each client ($c \in \{2, 3, 5, 7\}$), and then the prediction performance results in the encrypted-domain are compared with the results of the plain-domain.

*6.2. Experimental Results*

We first experimented with the COVID-19 detection model with no encryption and federated learning. In this case, there is only one client observed. Table 5 shows the model performance scores, and the running time was 599.169577s.

**Table 5.** Prediction result without federated learning.

| Precision | Recall | F1 Score | Accuracy |
|---|---|---|---|
| 0.868924 | 0.840000 | 0.836801 | 0.840000 |

We then applied federated learning in the model and observed the model performance based on evaluation matrices. Table 6 shows the performance results of federated learning without encryption.

**Table 6.** Performance measurements: federated learning without encryption.

| Number of Clients | 2 | 3 | 5 | 7 |
|---|---|---|---|---|
| Precision | 0.872128 | 0.865112 | 0.859288 | 0.850277 |
| Recall | 0.845000 | 0.837500 | 0.835000 | 0.827500 |
| F1 Score | 0.842123 | 0.834369 | 0.832164 | 0.824649 |
| Accuracy | 0.845000 | 0.837500 | 0.835000 | 0.827500 |

We then continued our observation by adjusting the hyperparameter *sec* between values 128 and 192.

Table 7 shows the performance measurements of federated learning with encryption level is 128.

**Table 7.** Performance measurements: federated learning with encryption *sec* = 128.

| Number of Clients | 2 | 3 | 5 | 7 |
|---|---|---|---|---|
| Precision | 0.867337 | 0.857293 | 0.853925 | 0.869584 |
| Recall | 0.837500 | 0.840000 | 0.830000 | 0.852500 |
| F1 Score | 0.834132 | 0.838040 | 0.827078 | 0.850776 |
| Accuracy | 0.837500 | 0.840000 | 0.830000 | 0.852500 |

Table 8 shows the performance measurements of federated learning with encryption level is 192.

**Table 8.** Performance measurements: federated learning with encryption *sec* = 192.

| Number of Clients | 2 | 3 | 5 | 7 |
|---|---|---|---|---|
| Precision | 0.866735 | 0.868924 | 0.855624 | 0.86800 |
| Recall | 0.840000 | 0.840000 | 0.832500 | 0.84500 |
| F1 Score | 0.837030 | 0.836801 | 0.829732 | 0.84254 |
| Accuracy | 0.840000 | 0.840000 | 0.832500 | 0.84500 |

Apart from measuring the model performance with evaluation matrices, we also measured total processing time, starting from model training at clients until the end of the federated learning process, and prediction result using the aggregated model. Table 9 shows the processing time in federated learning with various numbers of clients.

**Table 9.** Running time in seconds.

| Number of Clients | Without Encryption | Encryption (*sec* = 128) | Encryption (*sec* = 192) |
|---|---|---|---|
| 2 | 594.165448 | 4333.672333 | 4765.874634 |
| 3 | 647.963712 | 5124.841524 | 7504.239611 |
| 5 | 720.175786 | 6777.249099 | 10,518.012003 |
| 7 | 948.704833 | 9223.281346 | 13,277.904182 |

Below are histograms by evaluation matrices, accuracy, precision, recall, F1 score to visualize influence of encryption and encryption key length to model performance. Figure 5 the prediction performance score with different metrics.

From the histograms above, we see that homomorphic encryption and its key length variations do not have much influence to model performance.

We also used histogram to visualize running time growth against homomorphic encryption and its key length variations. From Figure 6, it shows that running time increased quite significantly with homomorphic encryption implemented. It also shows that running time was also influenced by encryption length—the longer the key, the longer the execution time.

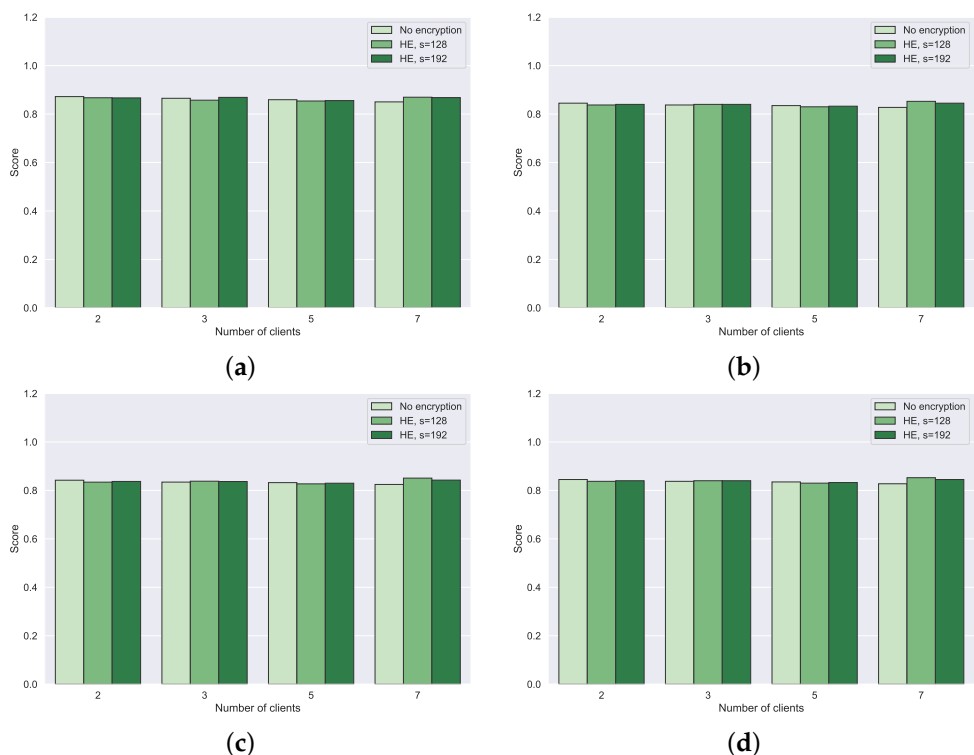

**Figure 5.** Prediction performance score with different metrics. (**a**) Precision. (**b**) Recall. (**c**) F1. (**d**) Accuracy.

One last thing we observed was that the pickle file size of encrypted weights was around 7 GB regardless of execution time.

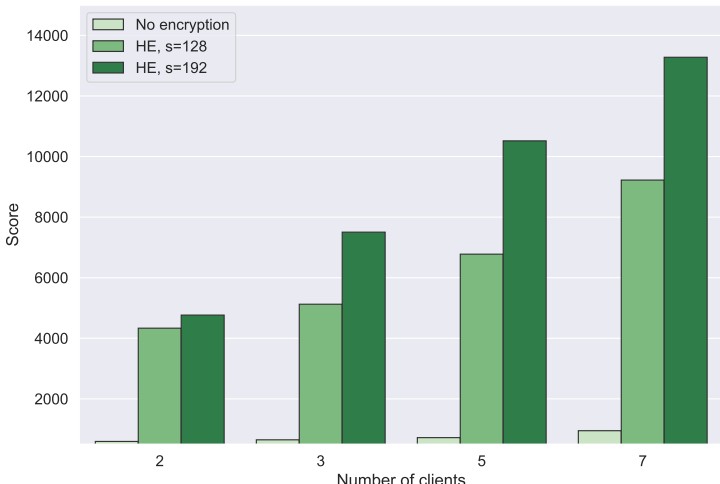

**Figure 6.** Running time in seconds.

## 7. Conclusions

Privacy-preserving has become an essential practice of healthcare institutions as both the EU, and we, mandate it. Federated learning and homomorphic encryption will play a critical role in maintaining data security and model training. By benefiting from both techniques, the proposed model achieves competitive performance while there is a significant trade-off for the execution time and the number of clients. In some cases, where privacy-preserving is very crucial, for instance, in healthcare fields, the trade-off is very acceptable.

The classification metrics, i.e., accuracy, F1, precision, and recall, reached over 80% using both encrypted and plain data for each federated learning case, which means that homomorphic encryption, in this case, SHE, does not deteriorate model performance.

Privacy attacks will cause immense damage to the security and privacy of patient information. This will hinder the advancement in healthcare using data-driven models. Therefore, it is indispensable to take crucial steps to strengthen the safety of the information, and the way data is processed. This study demonstrated that federated learning with homomorphic encryption could successfully enhance data-driven models by eliminating and minimizing the share of sensitive data. It is envisioned that this study could be helpful for the scientists and researchers working on sensitive healthcare data in multi-party computation settings.

**Author Contributions:** Conceptualization, F.W. and F.O.C.; methodology, F.W.; software, F.W.; validation, F.W. and F.O.C.; formal analysis, S.S. and M.K.; writin—original draft preparation, F.W. and F.O.C.; writing—review and editing, F.W., F.O.C., S.S. and M.K.; visualization, F.W.; supervision, F.O.C.; All authors have read and agreed to the published version of the manuscript.

**Funding:** This research received no external funding.

**Institutional Review Board Statement:** Not applicable.

**Informed Consent Statement:** Not applicable.

**Data Availability Statement:** The dataset used in this simulation is publicly avaliable COVID-19 dataset https://github.com/ieee8023/covid-chestxray-dataset accessd on 1 April 2022.

**Conflicts of Interest:** The authors declare no conflict of interest.

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
