# Peer review of "BFV-Based Homomorphic Encryption for Privacy-Preserving CNN Models"

_cryptography, doi:10.3390/cryptography6030034_

Round 1

Reviewer 1 Report

This article considers technique that uses a secure multi-party computation protocol to safeguard the deep learning model from adversaries. Authors use a homomorfic encryption for data privacy-preserving. Article is wel-wriiten and well-organised. Also, it sounds very scientific.

Authors should verify the text because there are some mistakes. For example, the sentence "Using homomorphic encryption, this research presents a privacy-preserving federated learning system for medical data." is two times.

Also, Figure 1 should be improved, because I can not read it all.

After these corrections, article can ce published.

Author Response

Dear Editor and Reviewers,
We are pleased to submit the revised version of the paper with the title “BFV based Homomorphic Encryption for Privacy-Preserving CNN models” for consideration in the Cryptography journal. We would like to thank the reviewers for their valuable comments on the paper. The provided comments have helped us to significantly improve the quality of the paper.

We have addressed all of the comments. For each comment, we provide a response on how we addressed it together with the key changes and references to the sections in the paper.

We have also colored the changes in red in our revised manuscript to ease the reviewing process.
The key changes are;
● We updated Figure 1.
● “Contributions” part is added to Introduction section.
● We improved the writing of the paper from the beginning to the end.
We look forward to hearing from you in due time regarding our submission and to respond to any further questions and comments you may have.

Comment 1. This article considers technique that uses a secure multi-party computation protocol to safeguard the deep learning model from adversaries. Authors use a homomorfic encryption for data privacy-preserving. Article is wel-wriiten and well-organised. Also, it sounds very scientific.

Response 1: Thank you for your comments.

Comment 2. Authors should verify the text because there are some mistakes. For example, the sentence "Using homomorphic encryption, this research presents a privacy-preserving federated learning system for medical data." is two times.
Response 2: We improved the writing of the paper from the beginning to the end.

Comment 3. Also, Figure 1 should be improved, because I can not read it all.
Response 3: We updated the figure and increased the font size.

Reviewer 2 Report

This work makes use of Deep Learning techniques for medical data protection. Federated learning is used, where training data is distributed among numerous machines and the learning process is performed collaboratively. There are numerous privacy attacks on deep learning (DL) models that allow attackers to obtain sensitive information. The authors argue that the DL model should be protected from adversarial attacks, especially in medical data applications. To this end, they propose homomorphic encryption against the adversarial collaborator as an answer to this challenge. This paper presents a federated learning system that aims to preserve the privacy of medical data. The proposed technique employs a secure multi-party computation protocol to protect the deep learning model from adversaries. The proposed approach is tested in terms of model performance using a real-world medical dataset. real-world medical data.

The system Model is interesting, although Figure 1 showing the overall system overview is too small and it is difficult to see the legends that appear.
In this work, COVID-19 Radiography dataset collected in previous works related to COVID-19 detection model ‘[ 22 ], [23 ] was used. The dataset contains X-ray lung images with four different classifications which are COVID, Lung_OPACITY, Normal, and Viral Pneumonia. Authors have utilized two classifications which are COVID and Normal, focusing only in COVID-19 detection machine learning process. I think that for future works it is possible to explore some works as “Protective Effect of Melatonin Administration against SARS-CoV-2 Infection: A Systematic Review” and “Analysis of Different Melatonin Secretion Patterns in Children With Sleep Disorders: Melatonin Secretion Patterns in Children".

I recommend the paper for publication since its technically sounds and provides a significant contribution to the state of the art.

Author Response

Dear Editor and Reviewers,
We are pleased to submit the revised version of the paper with the title “BFV based Homomorphic Encryption for Privacy-Preserving CNN models” for consideration in the Cryptography journal. We would like to thank the reviewers for their valuable comments on the paper. The provided comments have helped us to significantly improve the quality of the paper.

We have addressed all of the comments. For each comment, we provide a response on how we addressed it together with the key changes and references to the sections in the paper.

We have also colored the changes in red in our revised manuscript to ease the reviewing process.
The key changes are;
● We updated Figure 1.
● “Contributions” part is added to Introduction section.
● We improved the writing of the paper from the beginning to the end.
We look forward to hearing from you in due time regarding our submission and to respond to any further questions and comments you may have.

Comment 1. This work makes use of Deep Learning techniques for medical data protection. Federated learning is used, where training data is distributed among numerous machines and the learning process is performed collaboratively. There are numerous privacy attacks on deep learning (DL) models that allow attackers to obtain sensitive information. The authors argue that the DL model should be protected from adversarial attacks, especially in medical data applications. To this end, they propose homomorphic encryption against the adversarial
collaborator as an answer to this challenge. This paper presents a federated learning system that aims to preserve the privacy of medical data. The proposed technique employs a secure multi-party computation protocol to protect the deep learning model from adversaries. The proposed approach is tested in terms of model performance using a real-world medical dataset. real-world medical data.
Response 1: Thank you for your comments.
Comment 2. The system Model is interesting, although Figure 1 showing the overall system overview is too small and it is difficult to see the legends that appear.
Response 2: We updated the figure and increased the font size.
Comment 3. In this work, COVID-19 Radiography dataset collected in previous works related to COVID-19 detection model ‘[ 22 ], [23 ] was used. The dataset contains X-ray lung images with four different classifications which are COVID, Lung_OPACITY, Normal, and Viral Pneumonia. Authors have utilized two classifications which are COVID and Normal, focusing only in COVID-19 detection machine learning process. I think that for future works it is possible to explore some works as “Protective Effect of Melatonin Administration against SARS-CoV-2 Infection: A Systematic Review” and “Analysis of Different Melatonin Secretion Patterns in Children With Sleep Disorders: Melatonin Secretion Patterns in Children".
Response 3: We updated the manuscript and added following part:
We limited our work to binary classification. In our subsequent work, we plan to use multi-class approaches in the literature [7,8].

Comment 2. I recommend the paper for publication since its technically sounds and provides a significant contribution to the state of the art.
Response 2: Thank you for the comment.

Reviewer 3 Report

The contribution and novelty of this work is hard to see.  The authors don't do a great job of presenting their contributions and after reading the work, it seems like the contributions that can be found are rather small.

It seems like the primary intellectual contribution (the "idea" per se) is to use homomorphic encryption to implement a client - server based CNN.  The particular application in this work is a CNN to detect COVID-19 in X-ray images.  However, the actual design and implementation is fairly straight-forward.  An existing encryption scheme, BFV encryption, already supports addition and multiplication.  So, matrix-matrix multiplication can be achieved entirely in cipher-text by simply using BFV encryption.  Since a CNN is basically only matrix-matrix multiplications, every aspect of the the problem is solved in a straightforward way. Furthermore, existing libraries are used for the BFV code and for the machine learning code.  So, the implementation is not a contribution of this work either

It seems that the project does not incur any interesting research challenges or questions.

As I reach sections 5 and 6, the narrative seems to shift to focus on the effectiveness of the CNN to accurately detect COVID-19.  Since the CNN model is not a contribution of this paper, I don't see why it is evaluated so thoroughly.  The only element of Section 6 that I found relevant and interesting was Table 9; the running time.  Since the homomorphic encryption is much slower than operating on the data directly, it incurs 10 - 14x slowdown.  But, the authors don't do anything in their design or implementation to address the running time.  Personally I think that an interesting research challenge would be to improve the running time.  On the other hand, the system took at most ~13,200 seconds = ~3hrs, which I think is already a very reasonable time for a COVID-19 diagnosis of an X-ray image.

In summary, I question the novelty and significance of this work.

Further Minor Comments

I think the distinction between "partially" and "somewhat" homomorphic is unnecessarily complex and ultimately unnecessary.  There are only three categories of operations that are relevant: (1) addition (supported by the first three), (2) multiplication (supported by the last three) and (3) all other operations (not supported by any encryption scheme). Really the entire section could be summarized in a single sentence: "BFV is a homomorphic encryption scheme that supports both addition and multiplication of the ciphertext"

The format of the paper is a little bit odd beginning in Section 3.  The authors make heavy use of "Remark" headers to make various comments.  This format seems awkward and unnecessary.  I recommend the authors simply write prose in paragraphs like most other papers and most other sections of this paper.

The font in Figure 1 is too small.  A good rule of thumb is to try to ensure that the size of the font(s) in the figure is not smaller than the size of the font(s) in the rest of the paper.  I can zoom in on the PDF to read the figure, so maybe this comment is moot.

Section 7 should be proof-read for grammar.

Author Response

Dear Editor and Reviewers,
We are pleased to submit the revised version of the paper with the title “BFV based Homomorphic Encryption for Privacy-Preserving CNN models” for consideration in the Cryptography journal. We would like to thank the reviewers for their valuable comments on the paper. The provided comments have helped us to significantly improve the quality of the paper.

We have addressed all of the comments. For each comment, we provide a response on how we addressed it together with the key changes and references to the sections in the paper.

We have also colored the changes in red in our revised manuscript to ease the reviewing
process.
The key changes are;
● We updated Figure 1.
● “Contributions” part is added to Introduction section.
● We improved the writing of the paper from the beginning to the end.
We look forward to hearing from you in due time regarding our submission and to respond to any further questions and comments you may have.

Comment 1. The contribution and novelty of this work is hard to see. The authors don't do a great job of presenting their contributions and after reading the work, it seems like the contributions that can be found are rather small.
It seems like the primary intellectual contribution (the "idea" per se) is to use homomorphic encryption to implement a client - server based CNN. The particular application in this work is a CNN to detect COVID-19 in X-ray images. However, the actual design and implementation is fairly straight-forward. An existing encryption scheme, BFV encryption, already supports addition and multiplication. So, matrix-matrix multiplication can be achieved entirely in
cipher-text by simply using BFV encryption. Since a CNN is basically only matrix-matrix multiplications, every aspect of the the problem is solved in a straightforward way. Furthermore, existing libraries are used for the BFV code and for the machine learning code. So, the implementation is not a contribution of this work either

It seems that the project does not incur any interesting research challenges or questions.
As I reach sections 5 and 6, the narrative seems to shift to focus on the effectiveness of the CNN to accurately detect COVID-19. Since the CNN model is not a contribution of this paper, I don't see why it is evaluated so thoroughly. The only element of Section 6 that I found relevant and interesting was Table 9; the running time. Since the homomorphic encryption is much slower than operating on the data directly, it incurs 10 - 14x slowdown. But, the authors don't do
anything in their design or implementation to address the running time. Personally I think that an interesting research challenge would be to improve the running time. On the other hand, the system took at most ~13,200 seconds = ~3hrs, which I think is already a very reasonable time for a COVID-19 diagnosis of an X-ray image.
Response 1: Thank you for your comments. “Contributions” part is added to Introduction section.
The main contributions of this paper are as follows:
● A homomorphic encryption-based federated learning algorithm is proposed to protect the
confidentiality of the sensitive medical data.
● A secure multi-party computation protocol is proposed to protect the deep learning
models from the adversaries.
● A real-world medical dataset is used to evaluate the proposed algorithm. The
experimental results show that the proposed algorithm can protect the deep learning model from the adversaries.

Comment 2. Further Minor Comments
I think the distinction between "partially" and "somewhat" homomorphic is unnecessarily complex and ultimately unnecessary. There are only three categories of operations that are relevant: (1) addition (supported by the first three), (2) multiplication (supported by the last three) and (3) all other operations (not supported by any encryption scheme). Really the entire section could be summarized in a single sentence: "BFV is a homomorphic encryption scheme
that supports both addition and multiplication of the ciphertext"

The format of the paper is a little bit odd beginning in Section 3. The authors make heavy use of "Remark" headers to make various comments. This format seems awkward and unnecessary. I recommend the authors simply write prose in paragraphs like most other papers and most other sections of this paper.

Response 2: We want to keep this format as it is. In our opinion, this format creates a connection between sections.

Comment 3. The font in Figure 1 is too small. A good rule of thumb is to try to ensure that the size of the font(s) in the figure is not smaller than the size of the font(s) in the rest of the paper. I can zoom in on the PDF to read the figure, so maybe this comment is moot.

Response 3: We updated the figure and increased the font size.

Comment 4. Section 7 should be proof-read for grammar.
Response 4: We rephrased Section 7 and used Grammerly to fix typos.

Round 2

Reviewer 3 Report

The authors have revised their work to address my concerns.  Of all the points I raised in my original review of this work, all have been resolved by the authors' edits except one.  That is; the contributions that can be found are rather small. 

I think the authors' explicit listing of their contributions helps the reader find their contributions and generally understand their paper better.  I still feel the contributions are rather small, since they basically amount to a straightforward use of existing BFV and deep learning libraries.

Author Response

Our one of the main contributions is practical implementation. We added a new item to the Contribution part. The new contribution is as follows:

The main contributions of this paper are as follows:

  • A recent European Data Protection Board (EDPB) Public Consultation stated the use of Secure Multi-Party Computation as an additional measure to the General Data Protection Regulation’s (GDPR) Article 46 transfer tools. Here, we provide how to implement practically Secure Multi-Party Computation in Federated Learning to improve the privacy and security of medical data.
  • A homomorphic encryption-based federated learning algorithm is proposed to protect the confidentiality of the sensitive medical data.
  • A secure multi-party computation protocol is proposed to protect the deep learning models from the adversaries.
  • A real-world medical dataset is used to evaluate the proposed algorithm. The experimental results show that the proposed algorithm can protect the deep learning model from the adversaries.
